# Visible light-driven C−H activation and C–C coupling of methanol into ethylene glycol

Shunji Xie[1], Zebin Shen[1], Jiao Deng[2], Pu Guo[1], Qinghong Zhang[1], Haikun Zhang[1], Chao Ma[3], Zheng Jiang [4], Jun Cheng [1], Dehui Deng[2] & Ye Wang [1]

The development of new methods for the direct transformation of methanol into two or multi-carbon compounds via controlled carbon–carbon coupling is a highly attractive but challenging goal. Here, we report the first visible-light-driven dehydrogenative coupling of methanol into ethylene glycol, an important chemical currently produced from petroleum. Ethylene glycol is formed with 90% selectivity and high efficiency, together with hydrogen over a molybdenum disulfide nanofoam-modified cadmium sulfide nanorod catalyst. Mechanistic studies reveal a preferential activation of C−H bond instead of O−H bond in methanol by photoexcited holes on CdS via a concerted proton–electron transfer mechanism, forming a hydroxymethyl radical ($\cdot CH_2OH$) that can readily desorb from catalyst surfaces for subsequent coupling. This work not only offers an alternative nonpetroleum route for the synthesis of EG but also presents a unique visible-light-driven catalytic C−H activation with the hydroxyl group in the same molecule keeping intact.

[1] State Key Laboratory of Physical Chemistry of Solid Surfaces, Collaborative Innovation Center of Chemistry for Energy Materials, National Engineering Laboratory for Green Chemical Productions of Alcohols, Ethers and Esters, College of Chemistry and Chemical Engineering, Xiamen University, Xiamen 361005, China. [2] State Key Laboratory of Catalysis, Collaborative Innovation Center of Chemistry for Energy Materials, Dalian Institute of Chemical Physics, Chinese Academy of Sciences, Dalian 116023, China. [3] Center for High Resolution Electron Microscopy, College of Materials Science and Engineering, Hunan University, Changsha 410082, China. [4] Shanghai Synchrotron Radiation Facility, Shanghai Institute of Applied Physics, Chinese Academy of Sciences, Shanghai 201204, China. These authors contributed equally: Shunji Xie, Zebin Shen, Jiao Deng, Pu Guo. Correspondence and requests for materials should be addressed to J.C. (email: chengjun@xmu.edu.cn) or to D.D. (email: dhdeng@dicp.ac.cn) or to Y.W. (email: wangye@xmu.edu.cn)

Methanol can be derived from a variety of carbon resources, such as natural gas or shale gas, coal, biomass, and carbon dioxide, and is an abundant and renewable one-carbon ($C_1$) building block[1]. Many types of chemicals can be produced from methanol, and the carbon–carbon (C–C) bond formation is the most attractive and challenging reaction in methanol chemistry. Current conversions of methanol involving C–C bond formation are restricted to dehydrative oligomerizations such as methanol-to-olefin and methanol-to-gasoline processes, which show limited selectivity to a specific product[2], as well as methanol carbonylation[3]. The importance of methanol chemistry is increasing under the background of growing interest in the utilization of nonpetroleum carbon resources (in particular, shale gas) and carbon dioxide for sustainable production of chemicals. This trend has become a strong incentive to develop new methods or new routes for the transformation of methanol via C–C coupling with high selectivity.

Traditionally, the conversion of methanol usually involves the activation of its O–H or C–O bond. The system that can selectively activate the unreactive C–H bond of methanol with the hydroxyl group intact and form C–C bond is rare[4]. The preferential activation of inert sp[3] α-C–H bond in an alcohol without affecting the hydroxyl group is highly challenging in synthetic chemistry and is of high academic significance[5, 6].

Here, we present a visible-light-driven dehydrogenative coupling of methanol into ethylene glycol (EG) (Eq. 1), in which the hydroxyl group keeps intact. EG is an important chemical having a number of applications[7]. In particular, EG is widely used for the manufacture of polyesters, predominantly poly(ethylene terephthalate) (PET). The annual production of EG is >25 million metric tons and the demand for EG is expected to increase at a rate of 5% per year[8]. In the current industry, EG is primarily produced from petroleum-derived ethylene via epoxidation to ethylene oxide (EO) and the subsequent hydrolysis of EO. This multistep process suffers from low efficiency due to the low EO yield and high energy consumption[7]. The dehydrogenative coupling of methanol would offer a fascinating nonpetroleum route for sustainable production of EG.

$$2CH_3OH \rightarrow HOCH_2CH_2OH + H_2 \quad (1)$$

We report that CdS is a unique catalyst for the conversion of methanol to EG under visible-light irradiation. The modification of CdS nanorods with $MoS_2$ nanofoams further enhances the activity and EG selectivity. An EG selectivity of 90% can be obtained with a yield of 16% and a quantum yield of above 5.0%. We demonstrate that the preferential activation of C–H bond in methanol is driven by photoexcited holes via a concerted proton–electron transfer (CPET) mechanism on CdS surfaces, forming ·$CH_2OH$ radical as an intermediate for EG formation.

## Results

**Photocatalysts Efficient for Methanol Coupling to EG.** It is noteworthy that Eq. 1 cannot proceed via conventional thermo-catalysis because of the thermodynamic limitation (Supplementary Fig. 1). Solar-energy-driven photocatalysis is a promising strategy to realize the C–C coupling under mild conditions[9–11]. However, so far, the photocatalytic C–C coupling has been mainly limited to larger molecules such as 2,5-dihydrofuran[10]. Basically, a semiconductor with the conduction-band edge higher (more negative) than the $H_2O/H_2$ redox potential and the valence-band edge lower (more positive) than the EG/$CH_3OH$ redox potential may photocatalyze Eq. 1 (Supplementary Fig. 2). However, over most semiconductors investigated, instead of EG, HCHO was formed as a major carbon-based product (Table 1), suggesting that the O–H bond is easier to be activated. EG was formed on ZnS, but ZnS only worked under ultraviolet (UV) irradiation[12] due to its large bandgap energy (3.6 eV, corresponding to $\lambda = 345$ nm). We discovered that CdS, a semiconductor with a bandgap energy of 2.4 eV (corresponding to $\lambda = 518$ nm), catalyzed the formation of EG with better selectivity under visible light. The catalytic behavior of CdS depended on its morphology (Supplementary Table 1), and CdS nanorods exhibited the best performance for EG formation among CdS samples with different morphologies (Supplementary Fig. 3).

**Table 1 Catalytic performances of some typical semiconductors**

| Catalyst | Formation rate (mmol $g_{cat}^{-1}h^{-1}$) | | | | | | | $e^-/h^{+a}$ | Selectivity[b] (%) | | |
|---|---|---|---|---|---|---|---|---|---|---|---|
| | EG | HCHO | HCOOH | CO | $CO_2$ | $H_2$ | $CH_4$ | | EG | HCHO | HCOOH |
| *UV-Vis light* | | | | | | | | | | | |
| $TiO_2$ | 0 | 1.6 | 0.11 | 0.16 | 0.042 | 2.0 | 0.053 | 0.91 | 0 | 84 | 5.6 |
| ZnO | 0 | 3.0 | 0.038 | 0.23 | 0.028 | 3.1 | 0.14 | 0.90 | 0 | 91 | 1.2 |
| g-$C_3N_4$ | 0 | 0.79 | 0.33 | 0.11 | 0 | 1.5 | 0.039 | 0.92 | 0 | 64 | 27 |
| ZnS | 1.3 | 2.2 | 0.067 | 0.083 | 0 | 3.4 | 0.087 | 0.92 | 54 | 43 | 1.3 |
| *Visible light* | | | | | | | | | | | |
| ZnS | 0 | 0 | 0 | 0 | 0 | 0 | 0 | — | — | — | — |
| $Cu_2O$ | 0 | 0.46 | 0 | 0 | 0 | 0.42 | 0 | 0.91 | 0 | 100 | 0 |
| $Bi_2S_3$ | 0 | 0.13 | 0.017 | 0.023 | 0 | 0.19 | 0 | 0.91 | 0 | 77 | 10 |
| CuS | 0 | 0.11 | 0.013 | 0 | 0 | 0.13 | 0 | 1.0 | 0 | 89 | 11 |
| CdS particle | 0.28 | 0.40 | 0 | 0 | 0 | 0.65 | 0 | 0.95 | 58 | 42 | 0 |
| CdS rod | 0.46 | 0.38 | 0 | 0 | 0 | 0.75 | 0 | 0.90 | 71 | 29 | 0 |
| $MoS_2$ sheet/CdS[c,d] | 6.0 | 2.3 | 0 | 0 | 0 | 7.5 | 0 | 0.91 | 84 | 16 | 0 |
| $MoS_2$ foam/CdS[c,d] | 11 | 2.5 | 0 | 0 | 0 | 12 | 0 | 0.92 | 90 | 10 | 0 |
| $MoS_2$ sheet | 0 | 0 | 0 | 0 | 0 | 0 | 0 | — | — | — | — |
| $MoS_2$ foam | 0 | 0 | 0 | 0 | 0 | 0 | 0 | — | — | — | — |

Reaction conditions: solution, 76 wt% $CH_3OH$ + 24 wt% $H_2O$, 5.0 $cm^3$; atmosphere, $N_2$; light source, 300-W Xe lamp; UV-Vis light, $\lambda = 320$–780 nm; visible light, $\lambda = 420$–780 nm
[a] The ratio of electrons and holes consumed in product formation was calculated by the equation of $e^-/h^+ = [2 \times n(H_2) + 2 \times n(CH_4)]/[2 \times n(EG) + 2 \times n(HCHO) + 4 \times n(HCOOH) + 4 \times n(CO) + 6 \times n(CO_2)]$
[b] Selectivity was calculated on a molar carbon basis
[c] CdS without designation denotes the CdS nanorod
[d] Sheet: $MoS_2$ nanosheet with a content of 5.0 wt%; foam: $MoS_2$ nanofoam with a content of 5.0 wt%

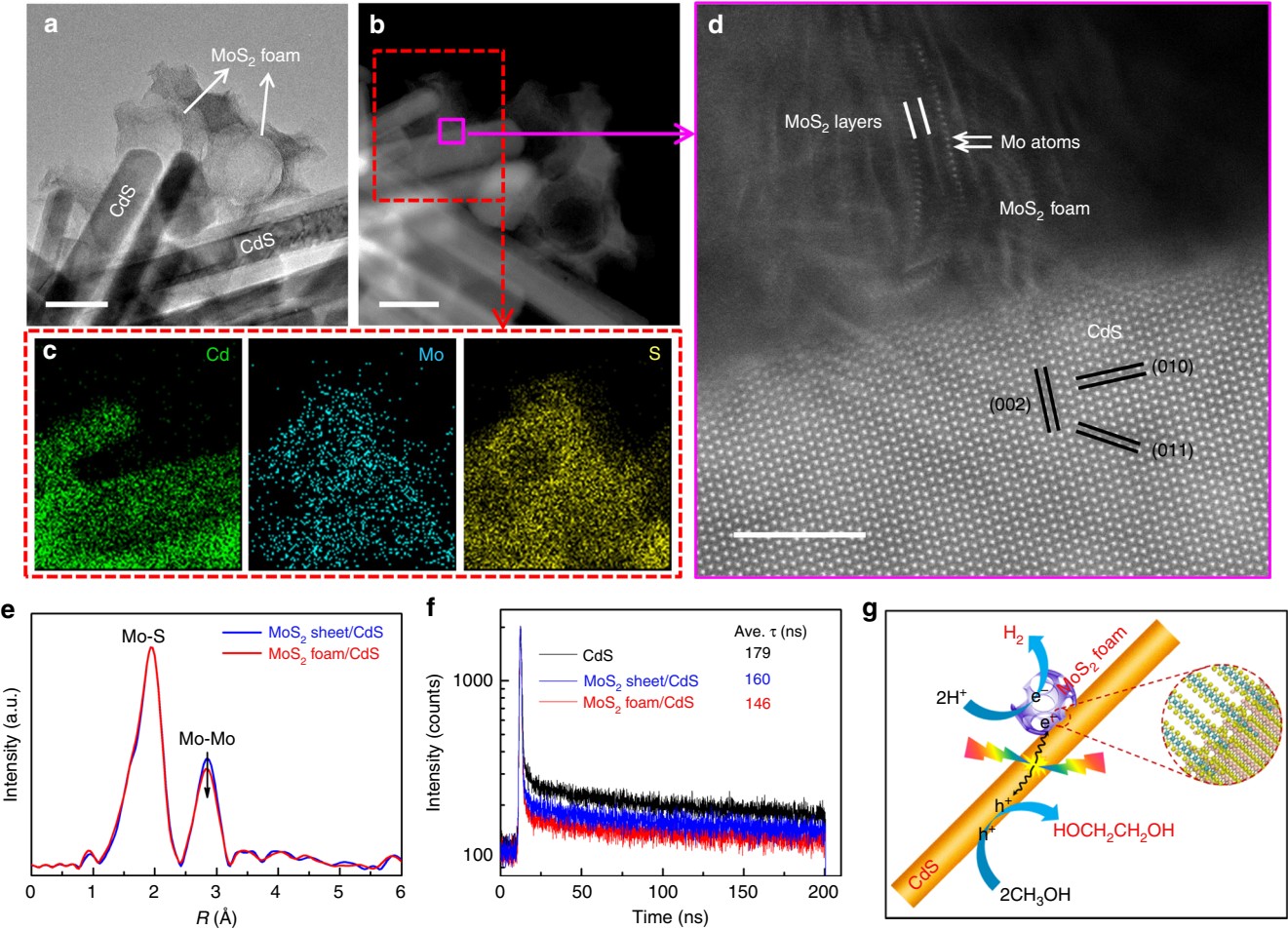

**Fig. 1** Structural and physicochemical properties of the MoS$_2$ foam/CdS catalyst. **a** TEM image of MoS$_2$ foam/CdS. **b** HAADF-STEM image of MoS$_2$ foam/CdS. **c** Corresponding EDX maps with a red rectangle in HAADF-STEM image of **b** showing the element distribution of Cd, Mo, and S. **d** High-resolution HAADF-STEM image of MoS$_2$ foam/CdS. **e** The $k^2$-weighted EXAFS spectrum of MoS$_2$ foam/CdS versus that of MoS$_2$ sheet/CdS. **f** Time-resolved photoluminescence (TRPL) spectra of CdS, MoS$_2$ sheet/CdS, and MoS$_2$ foam/CdS. **g** Schematic illustration of MoS$_2$ foam/CdS for photocatalytic synthesis of EG and H$_2$ from CH$_3$OH. Blue and red lines in **e** and **f** represent MoS$_2$ sheet/CdS and MoS$_2$ foam/CdS, respectively. The black line in **f** represents CdS. Scale bar: **a**, **b** 50 nm; **d** 5 nm

HCHO was a major by-product along with EG and H$_2$. We estimated the ratio of photogenerated electrons and holes consumed in product formation by assuming Eqs. 2–4, and the value was close to 1.0 for CdS. This confirms the occurrence of reactions of Eqs. 2–4.

$$2CH_3OH + 2h^+ \rightarrow HOCH_2CH_2OH + 2H^+, \qquad (2)$$

$$CH_3OH + 2h^+ \rightarrow HCHO + 2H^+, \qquad (3)$$

$$2H^+ + 2e^- \rightarrow H_2. \qquad (4)$$

**Superior Performances of MoS$_2$-Foam-Modified CdS Nanorods.** We loaded some typical co-catalysts[13] onto CdS nanorods to enhance the catalytic performance. Although all the co-catalysts investigated accelerated the formation of H$_2$, the formation of HCHO was enhanced more significantly than that of EG, leading to lower EG selectivity in most cases (Supplementary Table 2). It is quite unique that the addition of MoS$_2$ onto the CdS nanorod not only enhances the formation of H$_2$ and EG but also significantly increases the EG selectivity (Table 1). Moreover, we

found that MoS$_2$ nanofoam is a better co-catalyst for EG formation than MoS$_2$ nanosheet. The increase in the loading of MoS$_2$ nanofoam from 1.0 to 5.0 wt% gradually increased the rates of H$_2$ and EG formations, but did not significantly change the rate of HCHO formation (Supplementary Table 2). Thus, the EG selectivity increased upon increasing the MoS$_2$ loading amount to 5.0 wt%. A too higher MoS$_2$ loading (7.0 wt%) was unbeneficial to EG formation (Supplementary Table 2). The rates of EG and H$_2$ formations over the 5 wt% MoS$_2$ foam/CdS reached 11 and 12 mmol g$_{cat}^{-1}$ h$^{-1}$, respectively, which were about 24 and 16 times higher than those for the CdS nanorod alone. Further, the rate of EG formation for our MoS$_2$ foam/CdS catalyst under visible-light irradiation was about 1 order of magnitude higher than that for ZnS under UV-light irradiation. The EG selectivity was 90% over the MoS$_2$ foam/CdS catalyst (Table 1), which was also significantly higher than that over CdS alone or ZnS.

We performed characterizations to understand the origin of the significant promoting effect of MoS$_2$ nanofoam. The high-resolution transmission electron microscopy (HRTEM) and high-angle annular dark-field scanning TEM (HAADF-STEM) studies revealed that the MoS$_2$ foam located on the CdS rod had more edge sites and more intimate contact with CdS than the MoS$_2$ sheet on CdS (Fig. 1a–1d, Supplementary Figs. 4, 5, and 6). The

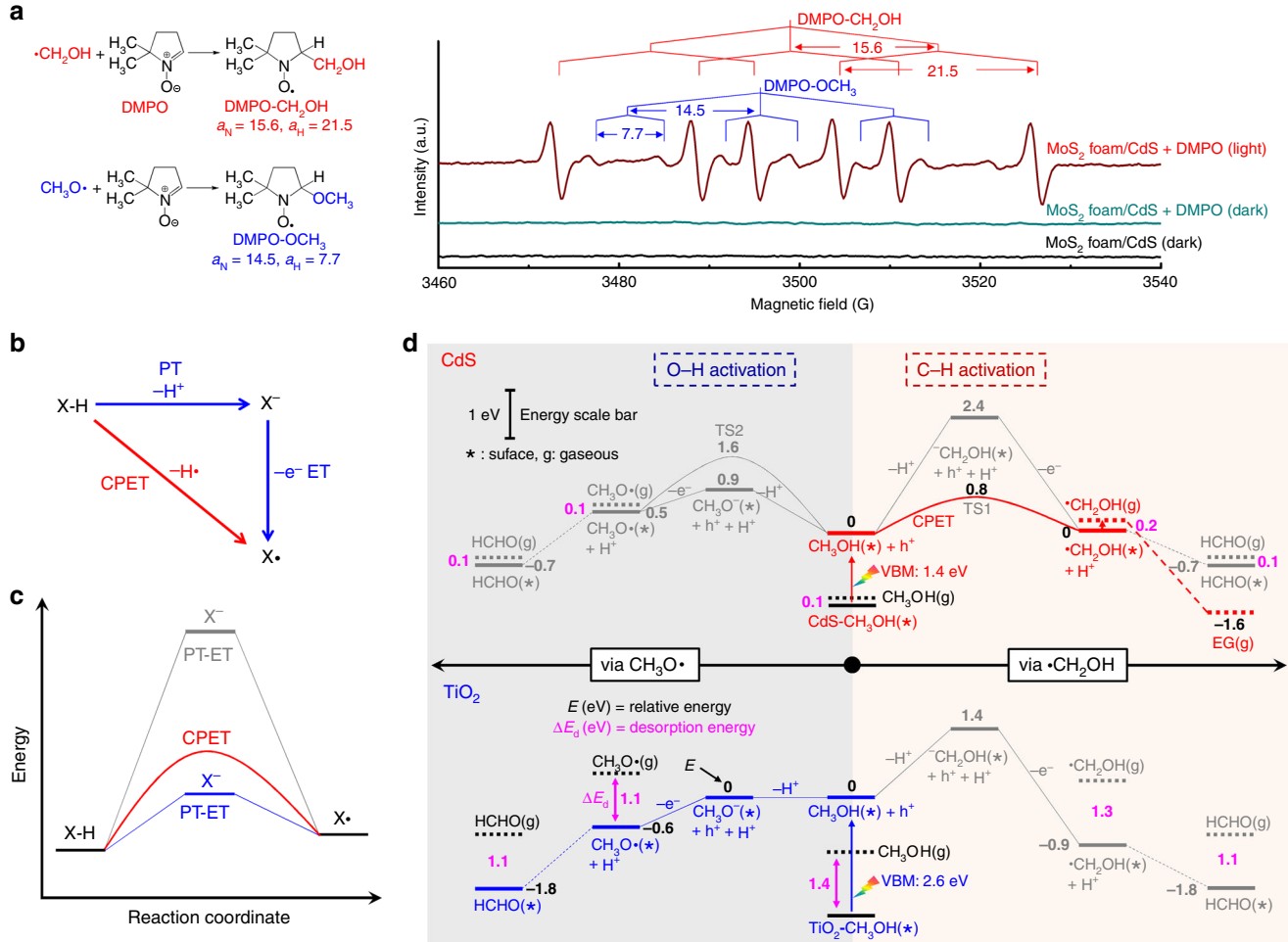

**Fig. 2** Mechanistic insights. **a** In situ ESR spectra for systems containing MoS$_2$ foam/CdS catalyst in methanol aqueous solution in the presence of DMPO (a spin-trapping agent) with or without light irradiation. **b** Two possible reaction pathways. **c** Energy profile. The CPET path is favored when the stepwise PT–ET path involves a high-energy intermediate. **d** Reaction energy profiles via ·CH$_2$OH and CH$_3$O· on CdS(100) and rutile TiO$_2$(110)

extended X-ray absorption fine structure (EXAFS) studies clarified that the MoS$_2$ foam on CdS possessed less Mo–Mo coordination than the corresponding MoS$_2$ sheet (Fig. 1e), also suggesting that the catalyst with MoS$_2$ foam had more edge sites. The MoS$_2$ edge sites are believed to be active sites for H$_2$ evolution[14, 15]. Actually, the MoS$_2$ foam/CdS catalyst also displayed a higher H$_2$ formation rate than CdS and MoS$_2$ sheet/CdS when a hole scavenger (Na$_2$S/Na$_2$SO$_3$ or lactic acid) was used instead of CH$_3$OH (Supplementary Table 3). On the other hand, the intimate contact between the MoS$_2$ foam and CdS rod may accelerate the transfer of photogenerated charge carriers, which is also a key parameter determining the performance. The photoluminescence (PL) intensity of the emission band at ~520 nm due to the recombination of photogenerated electrons and holes decreased in the following sequence: CdS >MoS$_2$ sheet/CdS >MoS$_2$ foam/CdS (Supplementary Fig. 7). The PL average lifetime (ave. $\tau$) derived from the time-resolved PL (TRPL) spectroscopy[16] decreased in the same order (Fig. 1f). These results confirm the enhancement in the separation and transfer of photogenerated excitons by the presence of MoS$_2$, in particular, MoS$_2$ foam. The photocurrent density and the cathodic current density of linear sweep voltammetry (LSV) measurements provided further evidence for this (Supplementary Figs. 8 and 9). Therefore, MoS$_2$ accelerates the photocatalytic activity for EG formation by

both providing H$_2$-evolution active sites and enhancing the transfer of photogenerated electrons and holes (Fig. 1g).

Furthermore, it should be reminded that the increase in the loading of MoS$_2$ foam from 1.0 to 5.0 wt% increased the EG formation rate but did not significantly change the HCHO formation rate. The average pore size of MoS$_2$ nanofoam derived from N$_2$ physisorption is 26 nm (Supplementary Fig. 10), while the size of ·CH$_2$OH is only 0.29 nm. Thus, the ·CH$_2$OH radicals can easily diffuse into the mesopores of MoS$_2$ nanofoam. We speculate that the mesoporous structure of MoS$_2$ nanofoam may provide more probability for the coupling of the reaction intermediate to form EG. Our kinetic measurements indicate that the formation of EG is a second-order reaction, whereas the formation of HCHO is a first-order reaction (Supplementary Fig. 11). The enrichment of the intermediate inside the mesopores of MoS$_2$ foam may be a reason for the enhancement in EG selectivity.

**Reaction Mechanism.** We performed deep studies to understand the reaction mechanism for EG formation. The addition of an electron scavenger (nitrobenzene) into the system with either CdS or MoS$_2$ foam/CdS catalyst stopped H$_2$ formation, whereas EG formation ceased and HCHO formation rate decreased drastically after the addition of a hole scavenger (Na$_2$S/Na$_2$SO$_3$)

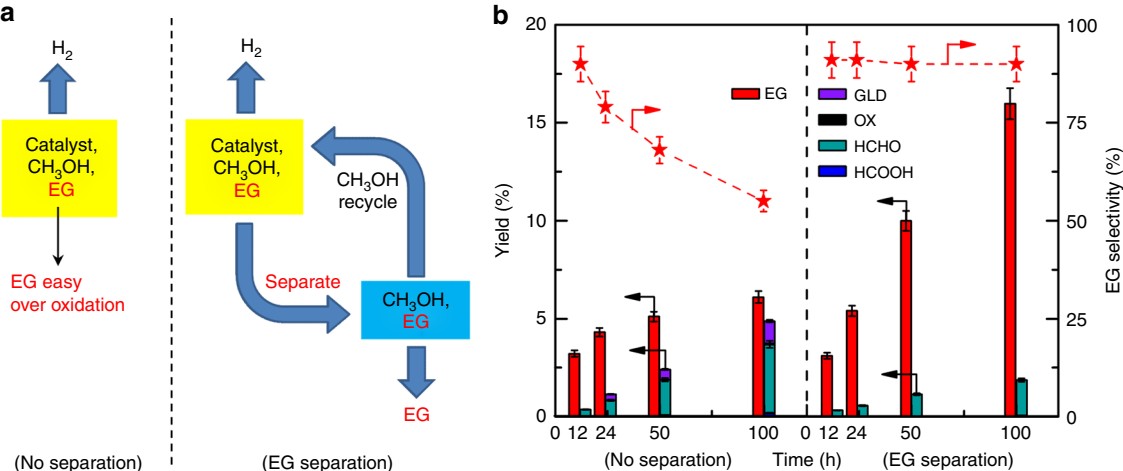

**Fig. 3** Process intensification. **a** Conventional reaction mode and process-intensified mode with EG separation. **b** Catalytic performance of MoS$_2$ foam/CdS. Reaction conditions: catalyst, 20 mg; solution, 76 wt% CH$_3$OH + 24 wt% H$_2$O, 10 cm$^3$; atmosphere, N$_2$; and light source, 300-W Xe lamp, visible light ($\lambda$ = 420–780 nm). The red star in **b** denotes EG selectivity. The experiments in each case were performed at least three times. The error bar represents the relative deviation, which is within 5%. GLD glycoaldehyde, OX oxalic acid

(Supplementary Table 4). These results provide further evidence for the reactions of Eqs. 2–4. The addition of a radical scavenger, 5,5-dimethyl-1-pyrroline-$N$-oxide (DMPO), also significantly suppressed the formation of EG and HCHO (Supplementary Table 4), suggesting that the formations of EG and HCHO proceed via radical intermediates. In situ electron spin resonance (ESR) spectroscopic studies using DMPO as a spin-trapping agent revealed the generation of the hydroxymethyl radical (·CH$_2$OH) and methoxyl radical (CH$_3$O·) on CdS and MoS$_2$ foam/CdS catalysts (Supplementary Fig. 12 and Fig. 2a). The ·CH$_2$OH radical should be an intermediate by C−H activation for the formation of EG, whereas the CH$_3$O· radical resulting from O−H activation may be responsible for HCHO formation.

What controls the preferential activation of C−H or O−H bond in methanol and the formation of intermediates over different catalysts is of paramount importance in mechanisms. To gain further insight into the reaction intermediate and the bond-activation mode, we performed density functional theory (DFT) calculations for methanol transformations on CdS and TiO$_2$, two semiconductors with different product distributions; the former provided EG as a major product, while the latter predominantly catalyzed the formation of HCHO without EG (Table 1). Generally, the formation of radical intermediates through X−H (X = C or O) cleavage by accepting a hole may proceed via either a stepwise pathway, i.e., proton transfer followed by electron transfer (PT−ET) or a CPET pathway, in which proton/electron transfer takes place in a concerted manner (Fig. 2b, c). On TiO$_2$ surfaces, the OH group of CH$_3$OH has a moderate deprotonation energy due to the strong interaction with the ionic oxide surfaces consistent with previous studies[17]. It is therefore more likely to take a two-step PT−ET route via CH$_3$O$^-$ to CH$_3$O·, eventually leading to the formation of HCHO (Fig. 2d, TiO$_2$). Such a route has been demonstrated in several previous studies[18–20]. On the other hand, CH$_3$OH has a negligible adsorption energy (−0.1 eV) on CdS, in contrast to that on TiO$_2$ (−1.4 eV). This indicates that the OH group of CH$_3$OH on CdS is difficult to deprotonate. The direct deprotonation of the C−H bond of methanol is even more difficult. It is often believed that the CPET takes place to avoid high-energy or highly unstable intermediates (Fig. 2c)[21, 22]. Our DFT studies suggest that the cleavage of the C−H bond in methanol occurs preferentially on CdS surfaces because

of the following reasons. First, the formation energy of ·CH$_2$OH is about 0.5 eV lower than that of CH$_3$O· (Fig. 2d, CdS). Second, assuming the nearby surface sulfur atom as a proton acceptor, the CPET driven by a hole state for the production of ·CH$_2$OH possesses a much smaller reaction barrier (0.8 eV) than that of CH$_3$O· (1.6 eV) (Fig. 2d, CdS). The formed ·CH$_2$OH intermediate has a small adsorption energy of −0.2 eV on CdS, and thus can readily desorb from the CdS surface, undertaking a thermodynamically downhill coupling to produce EG. We also found that ZnS, showing 54% EG selectivity, weakly binds ·CH$_2$OH with an adsorption energy of −0.5 eV (Supplementary Table 5). Hence, we believe that the weak adsorption of ·CH$_2$OH on catalyst surfaces plays a key role in the formation of EG. The strong adsorption of ·CH$_2$OH, even if produced, on TiO$_2$ (adsorption energy, −1.3 eV) and CuS (adsorption energy, −1.0 eV) will keep the intermediate on the surfaces, which then undergoes consecutive oxidation to form products such as HCHO (Fig. 2d, TiO$_2$ and Supplementary Table 5).

**Process Intensification and Quantum Yield.** Considering that EG may undergo consecutive oxidation in the reaction system, we have designed a process-intensified reactor that can perform simultaneous EG separation during the reaction (Fig. 3a, Supplementary Figs. 13 and 14). In this reactor with EG separation, the MoS$_2$ foam/CdS catalyst could be easily recovered and used repeatedly without significant deactivation (Supplementary Fig. 15). As compared with the conventional reaction mode, the process-intensified mode demonstrated a high EG selectivity (90%) during the longtime reaction. On the contrary, EG selectivity decreased significantly with reaction time, and many by-products such as glycoaldehyde, oxalic acid, HCHO, and HCOOH were observed in the conventional reactor (Fig. 3b, Supplementary Table 6). Therefore, EG yield could reach as high as 16% after 100 h in the reactor with EG separation.

We have measured the apparent quantum yields of EG under irradiation with different wavelengths. The quantum yield of EG was above 5.0% at wavelengths not longer than 450 nm for the MoS$_2$ foam/CdS catalyst, and decreased upon increasing the wavelength (Supplementary Fig. 16). The longest wavelength suitable for EG formation (~500 nm) was found to coincide with the absorption edge of the MoS$_2$ foam/CdS catalyst, which

was obtained from the diffuse reflectance UV-Vis measurement. This further indicates that the EG formation is indeed driven by light.

## Discussion

The present CdS-based photocatalytic system is quite unique in the preferential activation of the C−H bond in methanol without affecting the O−H group, forming EG via ·CH$_2$OH radical intermediate. This is achieved by photoexcited holes via the CPET mechanism on CdS surfaces. The weak adsorption of CH$_3$OH and ·CH$_2$OH intermediate on CdS decreases the possibility of O−H bond activation, which is a case on TiO$_2$, and enables the facile desorption of ·CH$_2$OH from catalyst surfaces for subsequent C−C coupling. The loading of MoS$_2$ nanofoam with abundant edge sites significantly improves the formation of H$_2$ and the overall activity. The EG selectivity is also enhanced probably because of the enriching effect of mesoporous nanofoam. The high selectivity of EG (90%) can be sustained in the long-term reaction by using a process-intensified reactor with EG-separation capability. The present visible-light-driven methanol transformation not only offers an atom-efficient method for the synthesis of EG under mild conditions, but also opens up a new avenue for preferential C–H bond activation without affecting other functional groups in the same molecule.

## Methods

**Synthesis of CdS Nanorods**. CdS nanorods were synthesized by a modified solvothermal method[23]. Typically, 4.62 g of CdCl$_2$·2.5H$_2$O and 4.62 g of CH$_4$N$_2$S were dissolved in 60 mL of ethylenediamine. Then, the mixture was transferred to a Teflon-lined autoclave and was maintained at 160 °C for >24 h. After cooling down to room temperature, the resulting yellow solid products were collected by centrifugation, and washed with distilled water and ethanol three times. The product was then dried at 60 °C.

**Synthesis of MoS$_2$ Nanofoam and Nanosheet**. MoS$_2$ nanofoam and nanosheet were synthesized by procedures reported previously[15]. For the synthesis of nanofoam, 0.4 g of (NH$_4$)$_6$Mo$_7$O$_{24}$·4H$_2$O and 1.6 g of SiO$_2$ nanospheres (30 wt% SiO$_2$ in EG, Alfa Aesar) were first dispersed in 20 mL of deionized water. After removing the solvent, the obtained powder reacted with 0.8 g of CH$_4$N$_2$S at 400 °C for 4 h. The obtained product was treated in hydrofluoric acid aqueous solution under room temperature, followed by washing with deionized water and drying. For the synthesis of the nanosheet, 0.9 g of (NH$_4$)$_6$Mo$_7$O$_{24}$·4H$_2$O was dissolved in 20 mL of deionized water and was then reacted with 10 mL of carbon disulfide at 400 °C for 4 h. The final product was obtained by treating in saturated NaOH aqueous solution at 60 °C, followed by washing with deionized water and drying.

**Preparation of MoS$_2$ Foam/CdS and MoS$_2$ Sheet/CdS Catalysts**. A series of MoS$_2$/CdS nanocomposites were prepared by an ultrasonic method[24]. For example, for the preparation of the MoS$_2$ foam/CdS catalyst, MoS$_2$ nanofoam (5.0 mg) was first ultrasonically dispersed in N,N-dimethylformamide (DMF,10 mL) in a flask for 3 h at room temperature. Then, CdS nanorods (100 mg) were added to the suspension. The mixture was further subjected to ultrasonic treatment for another 2 h to achieve close contact between MoS$_2$ and CdS. The MoS$_2$/CdS nanocomposite was collected by centrifugation and washed with deionized water and ethanol, followed by drying at 60 °C.

**Catalytic Reaction**. Photocatalytic reactions were carried out in a sealed quartz-tube reactor (volume, 20 mL). The visible (Vis) light source was a 300-W Xe lamp with a UV cutoff filter (420–780 nm). The UV-Vis light (320–780 nm) irradiation without using the UV cutoff filter was also applied to some catalysts. The solid catalyst powder (10 mg) was ultrasonically dispersed in 5.0 mL of mixed solution containing 76 wt% CH$_3$OH and 24 wt% H$_2$O. Then, the reactor was evacuated and filled with high-purity (99.999%) nitrogen. The photocatalytic reaction was carried out at room temperature typically for 12 h. After the reaction, the liquid products were analyzed by high-performance liquid chromatography (HPLC, Shimadzu LC-20A) with refractive index and UV detectors together with a Shodex SUGARSH-1011 column (8 × 300 mm) using a dilute H$_2$SO$_4$ aqueous solution as the mobile phase. H$_2$, CH$_4$, CO, and CO$_2$ were analyzed by an Agilent Micro GC3000 equipped with a molecular sieve 5A column and a high-sensitivity thermal conductivity detector. The relative deviation of detection was 4% for gas chromatography and 3% for liquid chromatography.

We measured the apparent quantum yields by using light at different wavelengths for the photocatalytic conversion of CH$_3$OH to EG over the 5% MoS$_2$

foam/CdS catalyst. The apparent quantum yield ($\eta$) for the formation of EG was calculated using the following equation:

$$\eta = [2n(EG) \times N_A] / [I(Wcm^{-2}) \times S(cm^2) \times t(s)/E_\lambda(J)] \times 100\%, \qquad (5)$$

where $n(EG)$, $N_A$, $I$, $S$, and $t$ represent the molar amount of EG, Avogadro's constant, light intensity, irradiation area, and reaction time, respectively. $E_\lambda$ can be calculated using $hc/\lambda$ ($\lambda = 380, 420, 450, 475, 500, 550,$ or 600 nm). Photocatalytic reactions with EG-separation mode (Supplementary Fig. 13) were carried out in the reactor shown in Supplementary Fig. 14. The light source was a 300-W Xe lamp with a UV cutoff filter (420–780 nm). The solid catalyst (20 mg) was ultrasonically dispersed in 10 mL of mixed solution containing 76 wt% CH$_3$OH and 24 wt% H$_2$O. Then, the reactor was evacuated and filled with nitrogen. The photocatalytic reaction was performed at room temperature. After the reaction, the liquid and gaseous products were also analyzed by HPLC and Micro GC.

**Characterization**. The photocatalysts or photocatalytic systems were characterized by scanning electron microscopy, TEM, high-resolution HAADF-STEM, energy-dispersive X-ray spectroscopy mapping, three-dimensional tomography, steady-state and TRPL spectroscopy, EXAFS spectroscopy, ESR spectroscopy, LSV, N$_2$ physisorption, diffuse reflectance UV-Vis spectroscopy, and photoelectrochemical measurements. The details of these techniques were described in Supplementary Information.

The other experimental and computational methods are displayed in Supplementary Information.

**Data Availability**. The data that support the findings of this study are available from the corresponding authors upon a reasonable request.

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

## Acknowledgements

This work was supported by the National Key Research and Development Program of the Ministry of Science and Technology of China (nos. 2017YFB0602201, 2016YFA0204100, and 2016YFA0200200), and the National Natural Science Foundation of China (nos. 21690082, 91545203, 21373166, and 21503176), the Key Research Program of Frontier Sciences of the Chinese Academy of Sciences (no. QYZDB-SSW-JSC020). We thank staffs at the BL14W1 beamline of the Shanghai Synchrotron Radiation Facilities (SSRF) for assistance with the EXAFS measurements.

## Author Contributions

S.X. and Z.S. performed most of the experiments and analyzed the experimental data. J. D. synthesized MoS₂ nanofoam, and performed LSV, HRTEM, and HAADF-STEM characterizations together with C.M. P.G. performed computational studies and analyzed the computational data. Q.Z. analyzed all the data and co-wrote the paper. H.Z. conducted a part of catalytic tests. Z.J. conducted EXAFS measurements and analyzed the results. J.C. guided the computational work, analyzed all the data, and co-wrote the paper. D.D. analyzed all the data and co-wrote the paper. Y.W. designed and guided the study, and co-wrote the paper. All of the authors discussed the results and reviewed the manuscript.

## Additional information

**Competing Interests:** The authors declare no competing interests.

