## [Peer Review File(PDF 183 kb) · Nature Communications]

Reviewers' comments:

Reviewer #1 (Remarks to the Author):

The work submitted by the group of Prof Wang showed a selective photocatalyzed preparation of ethylene glycol starting from methanol. Two main advantages of this method were apparent. First the use of visible light to promote the process and second the high selectivity obtained thanks to the use of a Mo based cocatalyst. The cleavage of the C-H bond in MeOH is not new due to the easy formation of the corresponding carbon centered radical.

I was, however, intrigued by the pathways that lead to HCHO. The Authors claimed (calculated) that this compound arose from O-H activation followed by oxidation to an alkoxy radical (Figure 2). I wonder if the same compound may arise from an oxydation/deprotonation sequence of the resulting HOCH₂(dot). Accordingly, the ms may be suitable for publication when this issue will be clarified.

Reviewer #2 (Remarks to the Author):

The authors describe an original and very interesting visible-light driven homocoupling of methanol to produce ethylene glycol in up to 90% selectivity on a heterogeneous catalyst system consisting of molybdenum disulfide nanofoam-modified cadmium sulfide nanorods. They have conducted detailed mechanistic studies both experimentally and using DFT methods, which demonstrate the preferred activation of C-H bonds over O-H bonds in the system by photoexcited holes on the catalyst. The study is of broad interest to the chemical and chemical engineering communitites as the direct transformation of methanol into multicarbon species in a controlled fashion is very challenging. The unique visible-light driven catalytic C-H activation is also highly interesting at a fundamental level. Moreover, the work demonstrates a chemical route towards ethylene glycol that does not require petroleum.

The manuscript is generally well written, holds high scientific and graphical quality and have only minor spelling/grammar issues. The topic is timely, of high interest and the work will be widely read and cited.

Recommendation: Publish as is.

Reviewer #3 (Remarks to the Author):

Photocatalytic synthesis is an interesting and important aspect in semiconductor photocatalysis. The authors reported a visible-light-driven dehydrogenative coupling of methanol into ethylene glycol (EG) over nanofoam-modified cadmium sulfide nanorod catalyst. Meanwhile, a synthetic mechanism study was carried out. This work might be suitable for publication in Nature Communications if the authors can address the following concerns properly:

- 1) The authors “estimated the ratio of photogenerated electrons and holes consumed in product formation by assuming Eqs. 2-4”, Page 4; however, for most cases in Table 1, the estimated e^-/h^+ is smaller than 1 and even 0.87 for TiO₂. Proper explanation should be given out on this issue.
- 2) The authors claimed that “The enrichment of intermediate inside the mesopores of MoS₂ foam may be a reason for the enhancement of EG selectivity.”, Page 7. Please offer the pore size and distribution information of mesopores MoS₂ and make a comparison with the size of intermediate like CH₃O• to clarify this assumption.
- 3) Based on “assuming the nearby surface sulfur atom as a proton acceptor” and theoretical simulation, a conclusion that “weak adsorption of •CH₂OH on catalyst surfaces plays a key role in the formation of EG” was drawn. However, it seems that this assumption lacks generality. For example, the selectivity of HCHO over CuS reached 89% while the selectivity to EG is 0. Please give out proper explanation.
- 4) The photocatalytic oxidation of methanol is a different perspective of photocatalytic H₂ evolution with the presence of hole scavenger. So, it should be comparable. However, in Table 1, the H₂ evolution rates over some traditional semiconductor were relatively low, particular for TiO₂. How to comprehend these results in addition to the absence of Pt as cocatalyst.
- 5) Wavelength dependence of quantum yield in EG production should be given to further identify a photocatalytic reaction.

Responses to Reviewers

Response to Reviewer 1

General Comments: The work submitted by the group of Prof Wang showed a selective photocatalyzed preparation of ethylene glycol starting from methanol. Two main advantages of this method were apparent. First the use of visible light to promote the process and second the high selectivity obtained thanks to the use of a Mo based cocatalyst.

Reply: We thank the reviewer for the positive comment on our work.

Comment 1: The cleavage of the C-H bond in MeOH is not new due to the easy formation of the corresponding carbon centered radical. I was, however, intrigued by the pathways that lead to HCHO. The Authors claimed (calculated) that this compound arose from O-H activation followed by oxidation to an alkoxy radical (Figure 2). I wonder if the same compound may arise from an oxydation/deprotonation sequence of the resulting HOCH₂(dot). Accordingly, the ms may be suitable for publication when this issue will be clarified.

Reply and actions taken: We thank the reviewer for the comments. We would like first to point out that the C-H cleavage is not an easy task, and it is one of the most important challenges in catalysis. Perhaps even more challenging in our case is the selective C-H cleavage in CH₃OH over O-H breaking. Our work presents the first visible-light-driven photocatalytic process producing high-value EG from CH₃OH with good efficiency and high selectivity. Although the focus of this work is the formation of EG via coupling of •CH₂OH from CH₃OH, we agree with the reviewer that the mechanism of HCHO formation may not be simple. There is some evidence, both experimental and theoretical, in literature indicating that CH₃O• is first produced via O-H breaking, followed by C-H cleavage to give HCHO (please see for examples: *J. Phys. Chem. B* 2002, **106**, 9122; *J. Phys. Chem. Lett.* 2011, **2**, 2707; *J. Am. Chem. Soc.* 2012, **134**, 13366; *J. Am. Chem. Soc.* 2013, **135**, 574; *ACS Catal.* 2017, **7**, 2374). However, we do not exclude the possibility that HCHO may be produced by dehydrogenation of •CH₂OH, which is formed by the breaking of C-H bond of CH₃OH. After all, •CH₂OH is the key intermediate for EG formation on CdS. The fast coupling of •CH₂OH significantly suppresses the pathway of O-H breaking, giving rise to low selectivity to HCHO on CdS. We have modified the main text to clarify this point: “*The strong adsorption of •CH₂OH, even if produced, on TiO₂ (adsorption energy, -1.3 eV) and CuS (adsorption energy, -1.0 eV) will keep the intermediate on the surfaces, which then undergoes consecutive oxidation to produce, e.g. HCHO (Fig.*

2c-ii and Supplementary Table 5)” (please see from Page 8-Line 2 from bottom to Page 9-Line 2). The path for the formation of HCHO via $\bullet\text{CH}_2\text{OH}$ has also been added in Figure 2c (Please see Page 18, Figure 2c).

Response to Reviewer 2

General Comments: The authors describe an original and very interesting visible-light driven homocoupling of methanol to produce ethylene glycol in up to 90% selectivity on a heterogeneous catalyst system consisting of molybdenum disulfide nanofoam-modified cadmium sulfide nanorods. They have conducted detailed mechanistic studies both experimentally and using DFT methods, which demonstrate the preferred activation of C-H bonds over O-H bonds in the system by photoexcited holes on the catalyst. The study is of broad interest to the chemical and chemical engineering communities as the direct transformation of methanol into multicarbon species in a controlled fashion is very challenging. The unique visible-light driven catalytic C-H activation is also highly interesting at a fundamental level. Moreover, the work demonstrates a chemical route towards ethylene glycol that does not require petroleum.

The manuscript is generally well written, holds high scientific and graphical quality and have only minor spelling/grammar issues. The topic is timely, of high interest and the work will be widely read and cited.

Recommendation: Publish as is.

Reply: We appreciate the very positive evaluation from this reviewer.

Response to Reviewer 3

General Comments: Photocatalytic synthesis is an interesting and important aspect in semiconductor photocatalysis. The authors reported a visible-light-driven dehydrogenative coupling of methanol into ethylene glycol (EG) over nanofoam-modified cadmium sulfide nanorod catalyst. Meanwhile, a synthetic mechanism study was carried out. This work might be suitable for publication in Nature Communications if the authors can address the following concerns properly.

Reply: We thank the reviewer for the pertinent comments on our work. Our replies to the issues raised by this reviewer and the corresponding revisions are described as follows.

Comment 1: The authors “estimated the ratio of photogenerated electrons and holes consumed in product formation by assuming Eqs. 2-4”, Page 4; however, for most cases in Table 1, the estimated e⁻/h⁺ is smaller than 1 and even 0.87 for TiO₂. Proper explanation should be given out on this issue.

Reply and actions taken: We thank the reviewer for this comment. The calculated ratio of e⁻/h⁺ in Table 1 is indeed slightly less than 1.0. We believe that the detection errors of gas/liquid chromatography measurements and the trace amounts of other products are the two main reasons for this issue. The detection errors of gas/liquid chromatography were estimated by testing the standard samples for at least three times. The relative deviation of detection was 4% for gas chromatography and 3% for liquid chromatography. The amount of electrons was calculated from the amount of H₂ and CH₄ detected by gas chromatography, and the amount of holes was calculated by the amount of CO, CO₂ detected by gas chromatography and EG, HCHO, HCOOH measured by liquid chromatography. Thus, the overall relative deviation of the estimated e⁻/h⁺ in the detection may reach about 7%. In addition, we have observed the formation of very small amount of CH₄ under UV-vis light irradiation. The amount of CH₄ produced on the UV-light-responsive catalyst has been added into Table 1 in the revised manuscript. CH₄ should be formed from CH₃OH via two-electron reduction process. Thus, we have recalculated the ratio of e⁻/h⁺ using the following equation:

$$e^{-}/h^{+} = [2 \times n(H_2) + 2 \times n(CH_4)] / [2 \times n(EG) + 2 \times n(HCHO) + 4 \times n(HCOOH) + 4 \times n(CO) + 6 \times n(CO_2)]$$

The e⁻/h⁺ value calculated using this equation for each catalyst is now in the range of 0.90-1.0. Therefore, we think that the calculated ratio of electrons and holes consumed in product formation is reasonably accurate.

We have added the rate of CH₄ formation in Table 1 (*please see Page 20, Table I*). The following sentence has been added in the footnote of Table 1 to describe the calculation of e⁻/h⁺: “*The ratio of electrons and holes consumed in product formation was calculated by the equation of e⁻/h⁺ = [2 × n(H₂) + 2 × n(CH₄)] / [2 × n(EG) + 2 × n(HCHO) + 4 × n(HCOOH) + 4 × n(CO) + 6 × n(CO₂)]*” (*please see Page 20, Table I, footnote*). We have further added the following sentences in the revised manuscript to describe the relative errors: “*H₂, CH₄, CO and CO₂ were analyzed by an Agilent Micro GC3000 equipped with a molecular sieve 5A column and a high-sensitivity thermal conductivity detector. The relative deviation of detection was 4% for gas chromatography and 3% for liquid chromatography*” (*please see Page 11, Paragraph 2, the last two sentences*).

Comment 2: The authors claimed that “The enrichment of intermediate inside the

mesopores of MoS₂ foam may be a reason for the enhancement of EG selectivity.”, Page 7. Please offer the pore size and distribution information of mesopores MoS₂ and make a comparison with the size of intermediate like CH₃O• to clarify this assumption.

Reply and actions taken: We thank the reviewer for this useful suggestion. We have performed N₂ adsorption measurements to gain information on pore-size distribution for MoS₂ foam. The obtained pore-size distribution has been displayed in Supplementary Figure 10 in the revised Supplementary Information. The average pore size of MoS₂ nanofoam was evaluated to be 26 nm. The sizes of •CH₂OH and CH₃O• intermediates obtained from the DFT calculation model are 0.29 nm and 0.21 nm, respectively. Compared with the average pore size of MoS₂ nanofoam, the size of •CH₂OH is much smaller, and thus the •CH₂OH intermediate can easily diffuse into the mesopores of MoS₂ nanofoam. The enrichment of •CH₂OH radicals inside the mesopores of MoS₂-foam may increase the probability of coupling of •CH₂OH to form EG, giving higher EG selectivity.

We have added the following sentences in the revised manuscript: “*The average pore size of MoS₂ nanofoam derived from N₂ physisorption is 26 nm (Supplementary Fig. 10), while the size of •CH₂OH is only 0.29 nm. Thus, the •CH₂OH radicals can easily diffuse into the mesopores of MoS₂ nanofoam*” (*please see Page 6-last three lines*).

Comment 3: Based on “assuming the nearby surface sulfur atom as a proton acceptor” and theoretical simulation, a conclusion that “weak adsorption of •CH₂OH on catalyst surfaces plays a key role in the formation of EG” was drawn. However, it seems that this assumption lacks generality. For example, the selectivity of HCHO over CuS reached 89% while the selectivity to EG is 0. Please give out proper explanation.

Reply and actions taken: We thank the reviewer for this important comment. To better understand the effect of surface adsorption, CuS (with zero EG selectivity) and ZnS (with 54% EG selectivity) have also been chosen to calculate the adsorption energies of •CH₂OH on the surfaces. The adsorption energies of •CH₂OH and CH₃O• on CuS and ZnS surfaces have been added into the Supplementary Table 5 in the revised manuscript. On the CuS surface, the adsorption energies of •CH₂OH and CH₃O• are -1.0 eV and -1.7 eV, respectively. Similar to TiO₂, with such strong adsorption, •CH₂OH will readily undergo further oxidation to form HCHO instead of desorption and coupling to form EG. In contrast, the •CH₂OH and CH₃O• intermediates have lower adsorption energies of -0.5 eV and -0.6 eV, respectively, on ZnS. Thus, the •CH₂OH intermediate can desorb from the ZnS surface relatively more easily for further coupling, consistent with our finding that ZnS has a moderate

selectivity of EG. All these results indeed show that weak adsorption of $\bullet\text{CH}_2\text{OH}$ intermediate on catalyst surfaces plays a key role in the formation of EG.

We have slightly revised Supplementary Fig. 17 by moving the adsorption energies $\bullet\text{CH}_2\text{OH}$ and $\text{CH}_3\text{O}\bullet$ intermediates to a new Table (Supplementary Table 5) in the revised Supplementary Information. The adsorption energies of $\bullet\text{CH}_2\text{OH}$ and $\text{CH}_3\text{O}\bullet$ intermediates calculated on ZnS and CuS surfaces have been added in the Supplementary Table 5 in the revised manuscript (*please see the Supplementary Information, Page 15*). We have revised our manuscript to add these results and discussions: “We also found that ZnS, showing 54% EG selectivity, weakly binds $\bullet\text{CH}_2\text{OH}$ with an adsorption energy of -0.5 eV (Supplementary Table 5). Hence, we believe that the weak adsorption of $\bullet\text{CH}_2\text{OH}$ on catalyst surfaces plays a key role in the formation of EG. The strong adsorption of $\bullet\text{CH}_2\text{OH}$, even if produced, on TiO_2 (adsorption energy, -1.3 eV) and CuS (adsorption energy, -1.0 eV) will keep the intermediate on the surfaces, which then undergoes consecutive oxidation to produce, e.g. HCHO (Fig. 2c-ii and Supplementary Table 5)” (*please see from Page 8-Line 5 from bottom to Page 9-Line 2*).

Comment 4: The photocatalytic oxidation of methanol is a different perspective of photocatalytic H_2 evolution with the presence of hole scavenger. So, it should be comparable. However, in Table 1, the H_2 evolution rates over some traditional semiconductor were relatively low, particular for TiO_2 . How to comprehend these results in addition to the absence of Pt as cocatalyst.

Reply and actions taken: We thank the reviewer for this comment. The TiO_2 used in our work is commercial Degussa P25, which have been widely used in photocatalysis. The H_2 formation rates on TiO_2 (P25) using methanol as a scavenger in some recent work have been listed in **Table R1** and the values are in the range of 0.36-4.6 $\text{mmol g}_{\text{cat}}^{-1} \text{h}^{-1}$. The difference should arise from the different reaction devices and reaction conditions used for the reaction. The H_2 formation rate on TiO_2 (P25) in our work is 2.0 $\text{mmol g}_{\text{cat}}^{-1} \text{h}^{-1}$, which is comparable to the results in the literature.

Table R1. H_2 formation rate on TiO_2 (P25) using methanol as a scavenger

Reference	H_2 formation rate ($\text{mmol g}_{\text{cat}}^{-1} \text{h}^{-1}$)
Our work	2.0
J. Catal. 2010, 273 , 182	0.36
Int. J. Hydrogen Energy 2010, 35 , 3991	4.6
Int. J. Hydrogen Energy 2013, 38 , 10739	0.40
J. Catal. 2015, 329 , 499	1.4

Comment 5: Wavelength dependence of quantum yield in EG production should be given to further identify a photocatalytic reaction.

Reply and actions taken: We appreciate this constructive comment from the reviewer to help further improve the quality of our manuscript. We have measured the apparent quantum yields using irradiations with different wavelengths ($\lambda = 380, 420, 450, 475, 500, 550$ and 600 nm) for the photocatalytic conversion of CH_3OH to EG over the 5% MoS_2 -foam/CdS catalyst. The results have been added in Supplementary Fig. 16 in the revised manuscript. The quantum yield of EG was above 5.0% at wavelengths ≤ 450 nm for the MoS_2 -foam/CdS catalyst, and decreased upon increasing the wavelength (Supplementary Fig. 16). The longest wavelength suitable for EG formation was found to coincide with the absorption edge of the MoS_2 -foam/CdS catalyst, which was obtained from the diffuse reflectance UV-vis measurement. This further indicates that the formation of EG is indeed driven by light.

Based on these results, we have added the following new paragraph in the revised manuscript: “*We have measured the apparent quantum yields of EG under irradiation with different wavelengths. The quantum yield of EG was above 5.0% at wavelengths not longer than 450 nm for the MoS_2 -foam/CdS catalyst, and decreased upon increasing the wavelength (Supplementary Fig. 16). The longest wavelength suitable for EG formation (~ 500 nm) was found to coincide with the absorption edge of the MoS_2 -foam/CdS catalyst, which was obtained from the diffuse reflectance UV-vis measurements. This further indicates that the EG formation is indeed driven by light.*” (**please see Page 9, Paragraph 3**)

Reviewers' comments:

Reviewer #1 (Remarks to the Author):

The modifications made by the Authors are fine for me. The ms may published as it is.

Reviewer #3 (Remarks to the Author):

The authors have addressed properly my concerns by providing additional evidence and discussion. I now recommend the acceptance of the paper.

Response to Reviewers

We thank the kind comments raised by the two reviewers. There are no revision requests now by the two reviewers.